# Lattice Structure and Spatial Network Models Incorporating into Simulating Human-Mediated Dispersal of the Western Conifer Seed Bug Populations in South Korea

Xiaodong Zhang [1,2], Dae-Seong Lee [3], Young-Seuk Park [3], Muyoung Heo [4], Il-Kyu Eom [1,5], Yang-Seop Bae [6], Tak-Gi Lee [6] and Tae-Soo Chon [2,5,7,*]

1. Department of Electronics Engineering, Pusan National University, Busan 46241, Republic of Korea; zxdytu1995@outlook.com (X.Z.); ikeom@pusan.ac.kr (I.-K.E.)
2. Ecology and Future Research Institute, Busan 46228, Republic of Korea
3. Department of Biology, College of Science, Kyung Hee University, Seoul 02447, Republic of Korea; dleotjd520@naver.com (D.-S.L.); parkys@khu.ac.kr (Y.-S.P.)
4. Research Solution Center, Institute for Basic Science, Daejeon 34126, Republic of Korea; mheo@ibs.re.kr
5. Research Institute of Computer, Information and Communication, Pusan National University, Busan 46241, Republic of Korea
6. Division of Life Sciences, Incheon National University, Incheon 22012, Republic of Korea
7. Industry and Business Foundation, Incheon National University, Incheon 22012, Republic of Korea
* Correspondence: tschon@pusan.ac.kr; Tel.: +82-10-4123-2261

**Abstract:** The western conifer seed bug (WCSB), *Leptoglossus occidentalis*, has expanded rapidly in the southern peninsula of Korea since it was first reported in southeastern Korea in 2010. Two types of human-mediated passive movements were devised for modeling the rapid advancement of the pest population in this study: traffic effects and forest-product transportation. A lattice structure model (LSM) was developed to accommodate the traffic effects pertaining to the local area along with the natural population dynamics of the pest. Separately, a spatial network model (SNM) was constructed to present the passive movement of the WCSB because of forest-product transportation between all local areas in Korea. The gravity rule was applied to obtain the parameters for forest-product transportation between the local areas. LSM and SNM were linked to the two present types of passive movements in the model. The model simulated fast, linear advancement in a short period, compared with slow, circular advancement because of the conventional natural diffusion process of populations. Simulation results were comparable to field data observed in the southern peninsula of Korea, matching the rapid advancement of about 400 km to the north area (Seoul) from the south area (Changwon) within six years and expanding across the nation in 10 years. Possible saturation of populations was predicted in the 2020s if survival conditions for the WCSB were favorable and no control efforts were given in field conditions. Dispersal because of SNM notably surpassed the dispersal simulated by LSM when the WCSB population rapidly dispersed over a wide area. The Allee-effect and contribution ratio of SNM were the factors governing the rapid expansion of pest populations. The possibility of using the combined model was further discussed to address different types of human-mediated passive movements associated with population dynamics in forest pest dispersal.

**Keywords:** forest pest; dispersal; passive movement; monitoring; pest management

## 1. Introduction

Owing to climate change and human aggregation, biological invasion has been a critical issue regarding population stability and biodiversity maintenance. Insect invasion has been noted because of serious damage to agriculture and forestry [1]. The damage caused by pest populations in forests is particularly serious because of the vulnerability of fast dispersal in spatially contiguous vegetation [2]. Since the western conifer seed bug (WCSB), *Leptoglossus occidentalis*, was reported in Changwon, Korea in 2010 [3], the

insect population has rapidly expanded and dispersed across the nation by 2019 [4]. WCSB is a species native to western North America and is considered a key pest of coniferous seed orchards [5]. Currently, WCSB is broadly dispersed across continents from North America to Asia and is mainly univoltine in temperate zones (with a possible variation in invading areas), with damage starting from nymphs' sucking leaf and cone nutrients after emergence [6,7]. By eating 1–2 seed cones per day [8], WCSB causes damage to host plants, including conelet abortion, seed fusion, and nutrient depletion [9], eventually resulting in germplasm reduction and natural regeneration [10]. Invasion by insects in Korea is associated with high-risk factors, including distance to the road (<500 m), followed by low altitude (<200 m), low precipitation (<180 mm for the wettest month), and distance to the forest (<100 m) [3,4,11]. Kim et al. [4] suggested that passive movement, because of sapling transportation, could cause the rapid dispersal of WCSB in Korea.

Mathematical modeling of invasive forest pests has been performed since the 1990s, initially with the diffusion model applied to the spongy moth *Lymantria dispar* [12]. Mathematical modeling of invasive processes in insects has been reviewed by Liebhold and Tobin [1], reporting a stratified dispersal model [13] for presenting one of the original long-distance movement models. Statistical and mechanistic models of insect invasion were reviewed by Vinatier et al. [14], who stated that statistical models were mainly applied to characterize spatial patterns. In contrast, mechanistic models were developed for presenting dispersal abilities according to behavior, individual interactions, and habitat selection.

Lee et al. [15] incorporated a lattice model with short and long movements to simulate the dispersal of forest pests, pine needle gall midge, and pine wilt disease, reporting the 'type b' dispersal pattern with a broken point in the pine needle gall midge as presented in Shigesada and Kawasaki [13]. Van Nguyen et al. [16] developed a spatially explicit model to present short- and long-distance dispersal in association with biological and environmental events concurrently, including the influence of neighboring infestation, asymptomatic carriers, and typhoon effects. The spatial conformations produced by the spatially explicit model effectively addressed the probability of disease occurrence from both local and global aspects.

Long-distance was also simulated using individual-based models (IBMs) for forest pest insects. Chon et al. [17] constructed an IBM for pine sawyer beetles with pine wilt disease. The difference in movement distances was simulated by Lèvy flight, presenting a long movement with a suitable parameter of 1.5. The model effectively forecasted the number and positions of infested lattices in a spatially explicit manner and estimated the influence of the selected parameters, including the control rates. Recently, Xia et al. [18] developed an IBM by combining the individual movement behavior of a forest pest, the pine sawyer beetle, with long- and middle-distance movements of individuals in step functions and provided practical information on the prognosis of population dynamics and management strategy. However, in these studies, simulations were conducted in association with natural population dynamics and human-mediated movements were not included to present rapid advancement of the pest population.

Regarding the human-mediated dispersal of invasive insects, Carrasco et al. [19] simulated long-distance dispersal along with the natural dispersal of the western corn root worm. The gravity rule was applied to domestic dispersal based on human population size, whereas crossing national boundaries (e.g., inland waters, motorways, and railways) was simulated according to the negative power law. Robinet et al. [20] modeled the short- and long-distance dispersal of a forest pest, the pinewood nematode. Although the reaction-diffusion model was used for short-distance dispersal, invasion probability was generated according to population size and potential anthropogenic pathways (e.g., rivers and lakes) in large scale. Robinet et al. [21] developed a similar model for a pine processionary moth with local spread based on a reaction-diffusion model and long-distance dispersal by generating long jumps randomly arriving in urban areas according to the host tree and human population densities.

Our model is similar to previous models in that short- and long-distance dispersals have different models, diffusion models for short dispersal, and some specialized models for long-distance dispersal, including the gravity rule. Our models differed in specifying different types of long-distance dispersal according to traffic load and forest-product transport. In addition, a spatial network model (SNM) was devised and combined with a lattice structure model (LSM). Natural population growth and passive movement because of traffic effects were presented using the LSM. Separately, the SNM was developed to present long-distance transport of WCSB owing to forest-product transportation between local areas. Subsequently, the SNMs and LSMs were linked in each iteration to combine all the active and passive movements of the WCSB. Model parameters were examined for fast dispersal of WCSB populations and simulation results were compared with field data.

## 2. Materials and Methods

### 2.1. Input Data and Parameters

#### 2.1.1. Field Data

Spatial dispersal data for the WCSB were obtained from the southern Korean peninsula [3,6,11,22] (Figure 1). The surveys were conducted nationally, reporting on-site observations of WCSB individuals annually since 2010. Notably, the population rapidly spread through the highway network, eventually leading to Seoul within 6 years (dotted ellipse in 2016, Figure 1).

#### 2.1.2. Transportation of Forest Products

As mentioned in [4], the enhancement of population advancement of WCSB could be caused by the indirect transportation of saplings between local areas. We assumed that sapling transportation is proportional to forest-product transportation in Korea. The forest-product transportation data between 16 metropolitan cities and provincial capitals in Korea were obtained on a large scale from the National Logistics Information Center (https://www.nlic.go.kr/nlic/frghtRoad0040.action; accessed on 6 April 2022). Since the data were only provided for large areas in Korea, we obtained transportation between small areas to present detailed transportation at local levels, including municipal cities and counties within provinces, 233 in total in Korea (Figure 2a). The gravity rule (listed below) [19,23] was applied to the input data in two steps to obtain the amount of transportation ($r_{ij}$) between all local areas in Korea in this study. First, the parameters in the gravity rule were obtained with the input data for large areas (population size and distances between areas), as stated above, and forest products (tons per year) transported between all large areas because the transportation data are only available for large areas in Korea. According to the least square method, the parameters for the gravity rule that were obtained were $\alpha$ = 0.64113291, $\gamma$ = −0.00877989, and $\varepsilon$ = 0.01037146. For the second step, we calculated the transportation between all local areas with the gravity rule using the obtained parameter values and data for local areas (population size and distances between all local areas) as input.

$$r_{ij} = \frac{H_i^\alpha H_j^\gamma}{e^{\varepsilon e_{ij}}} \qquad (1)$$

$H_i^\alpha$: Human population size of $i$ node with exponential $\alpha$.
$H_j^\gamma$: Human population size of $j$ node with exponential $\gamma$.
$e_{ij}$: Geographical distance (linear) from node $i$ to node $j$.
$\alpha$ and $\gamma$: Constants.
$\varepsilon$: Inverse characteristic distance.
$r_{ij}$: Transportation of forest products from node $i$ to node $j$.

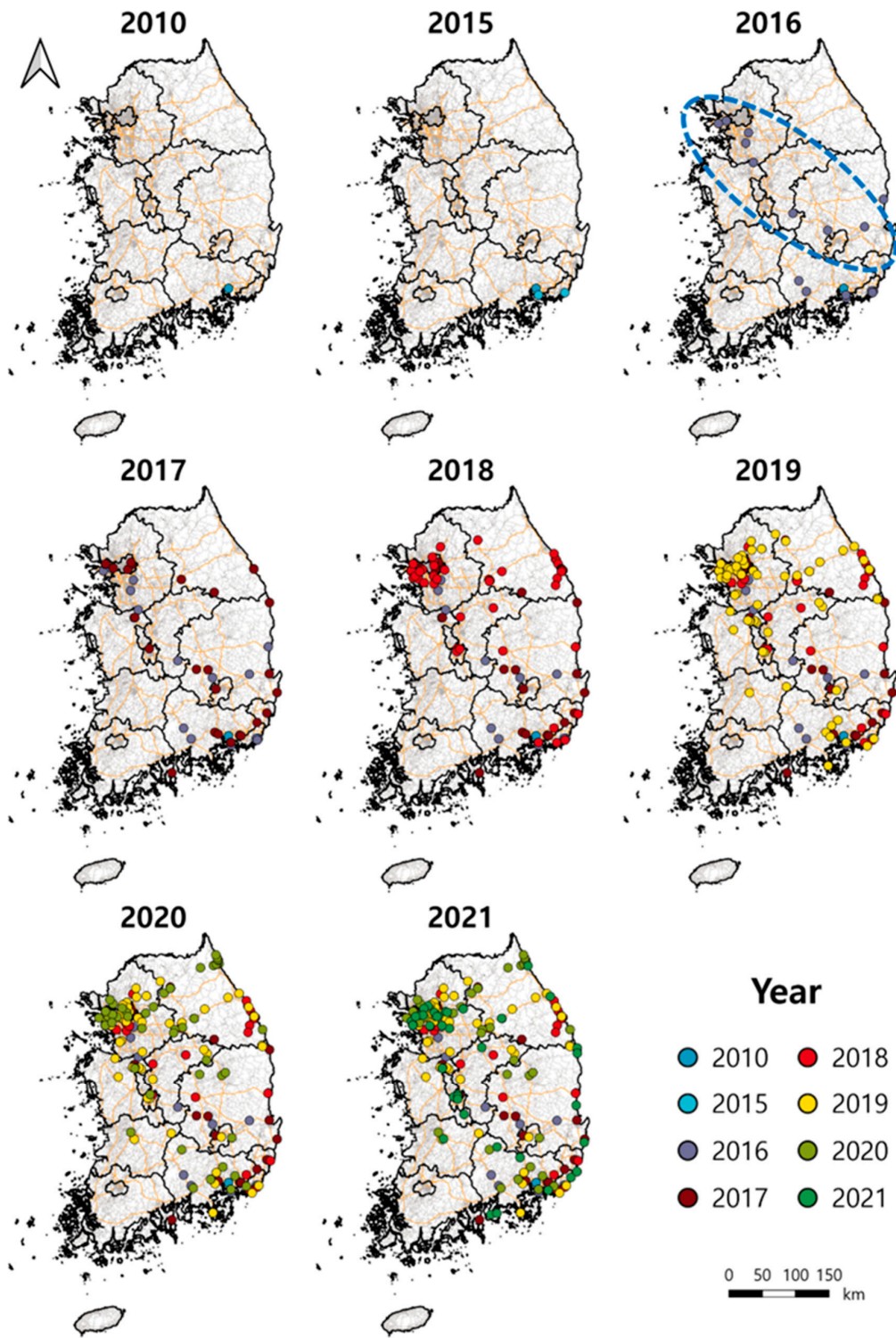

**Figure 1.** Dispersal of the WCSB populations in Korea between 2010–2021 [3,6,11,22] (data for 2019~2021 obtained from the field) (Orientation in all subfigures is the same shown in year 2010).

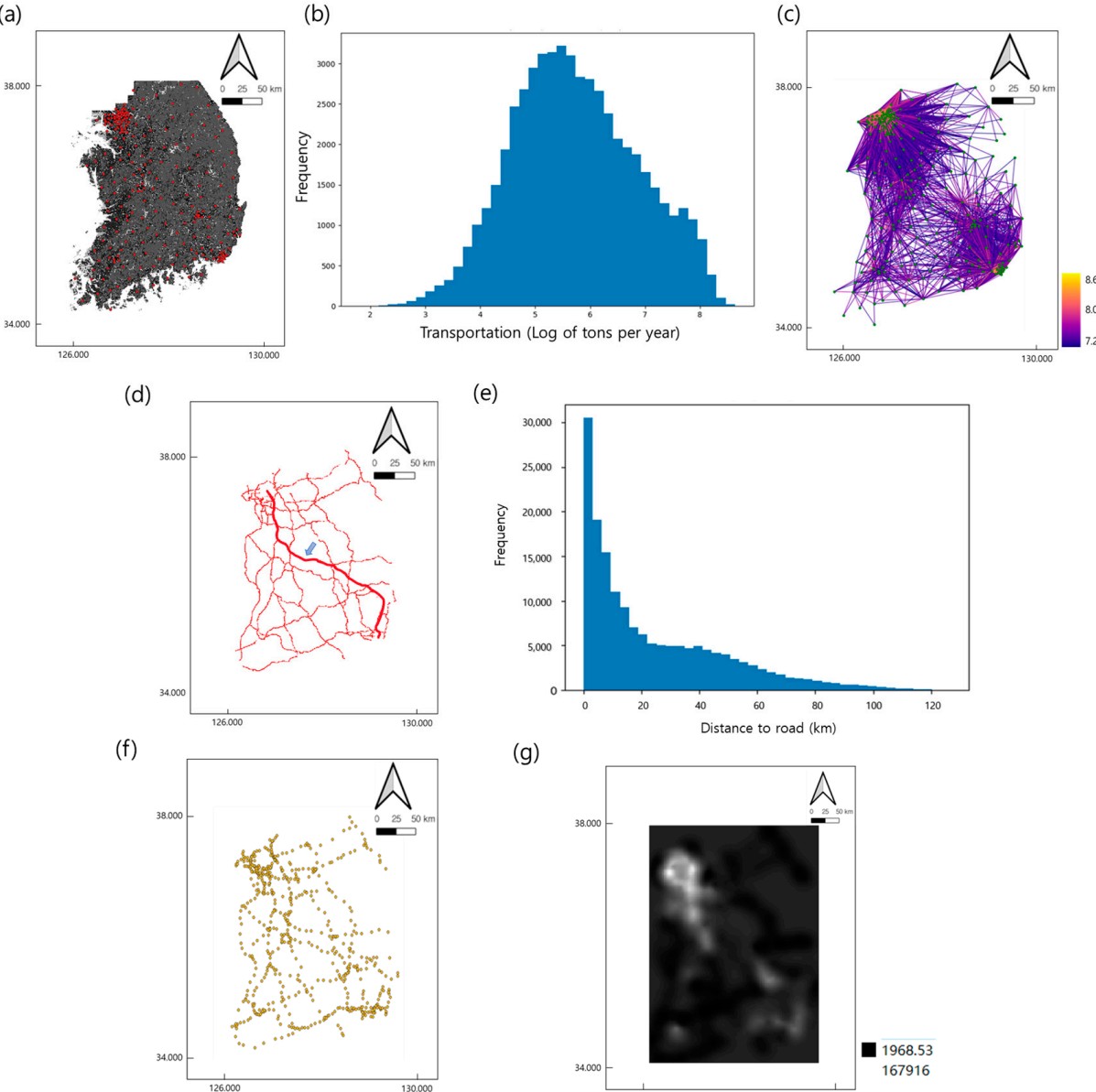

**Figure 2.** Position of capitals in municipal cities and counties within provinces in Korea (red dots) (**a**), frequency of edges (forest-product transportation between local areas) (**b**), edges over the threshold (7.0 in common log.) on the map (No. of nodes; 233, Threshold value; 7.0 (Log. of tons per year), and No. of edges; 8402) (green dots presenting local area capitols) (**c**), highway network in the southern peninsula of Korea (thick red line indicated by an arrow to present the Gyeongbu Highway from Busan to Seoul) (**d**), frequencies of distances from all spatial units to the nearest point of highway (**e**), and sampling points for traffic load (no. of cars per day) in Korea in the highway network (**f**) with kriging data (**g**).

The peak of the edges ($r_{ij}$; the amount of forest-product transport per year in tons) was observed with 5.5 tons (common log) per year (Figure 2b). The frequencies rapidly increased as transportation was close to 5.5 tons and decreased linearly beyond this point. Figure 2c shows the transportation ($r_{ij}$) between all local areas with edges greater than 7.0 (common log.) tons per year. Higher connections were observed mainly in the Seoul area (northwest), followed by the Busan area (southeast).

### 2.1.3. Closeness to the Highway Network

To estimate the degree of closeness to the highway, the distance from all spatial units to the highway was obtained. The highway network data were obtained from the website of the National Spatial Data Infrastructure Portal (http://data.nsdi.go.kr/dataset/201809 27ds0062; accessed on 10 April 2022) (Figure 2d). Highways in this study are defined as national, limited-access expressways in Korea where car speed is allowed 100–120 km per hour with a minimum road width of 3.5 m (Ordinance 922 (13 December 2021), Ministry of Land, Infrastructure, and Transport, Republic of Korea). The Gyeongbu Highway is the backbone road from Seoul to Busan, passing from north to south in Korea (arrow, Figure 2d). In Korea, road systems are composed of main arterial roads, auxiliary arterial roads, reception/distribution roads, and local roads, according to spatial scales. Highways defined in this study belong to the largest category on the main arterial road and serve as the major connecting network between large regional administrative units, including metropolitan cities (e.g., Seoul, Busan) and provinces (e.g., Gyeonggi-do, Gyeongsangnam-do). Since roads at smaller scales are mainly related to transportation within the regions (and local areas), and national-scale transportation should eventually pass through highways, we used highway networks to simulate the dispersal of WCSB on a national scale in Korea. The frequency of the distances from every cell in the simulation system to the nearest highway is presented in Figure 2e. A strong hollow curve was observed with the maximum level of minimum distances of 0.0–0.3 km, indicating most areas close to highways. The frequencies substantially decreased beyond 20 km.

### 2.1.4. Traffic Load

To estimate the traffic load, we used the data for the number of vehicles, including passenger cars, buses, small cargo, heavy cargo, and large cargo, as observed from the Transportation Monitoring System (https://www.road.re.kr/itms/itms_01.asp?pageNum= 3&subNum=2; accessed on 7 July 2022) in the highway network in Korea (623 observation points) (Figure 2f). An average of 60,593 vehicles were observed per day at each observation point. Kriging [24] was conducted to present traffic load in two dimensions (Figure 2g) to be used for input data for determining the distance of passive movements in the model (See Section 2.2.4). Heat maps are drawn based on the number of observation points and tend to be biased depending upon the number of sampling points (i.e., a higher number of sampling points tending to have higher values just because of the number of sampling points, not by the values observed at the sampling points). Kriging is conducted based on interpolation of actually measured values and minimizes prediction errors caused by differences in the sample numbers [25]. Since the observation points for traffic load are unevenly distributed and concentrated in the area of high human population size as stated above (Figure 2f), kriging was used to transform point data to spatial data in two dimensions in this study. The traffic load was outstandingly high surrounding the Seoul region in the northwest area, followed by the traffic load in the Busan area in the southeastern area, which was substantially lower than that in the Seoul region (Figure 2g).

### 2.1.5. Distribution of Sapling Farms

Data on the distribution of sapling farms in Korea were downloaded from the Kakao map (https://map.kakao.com/; accessed on 9 November 2022) and used as input to enhance forest-product diffusion within local areas in LSM. Figure 3a shows the actual distribution of sapling farms in Korea. A specific area in the middle of the southern peninsula in Okcheon-gun had an outstandingly higher number of farms. Figure 3b shows a heat map of the sapling farms to provide 2D data for the model.

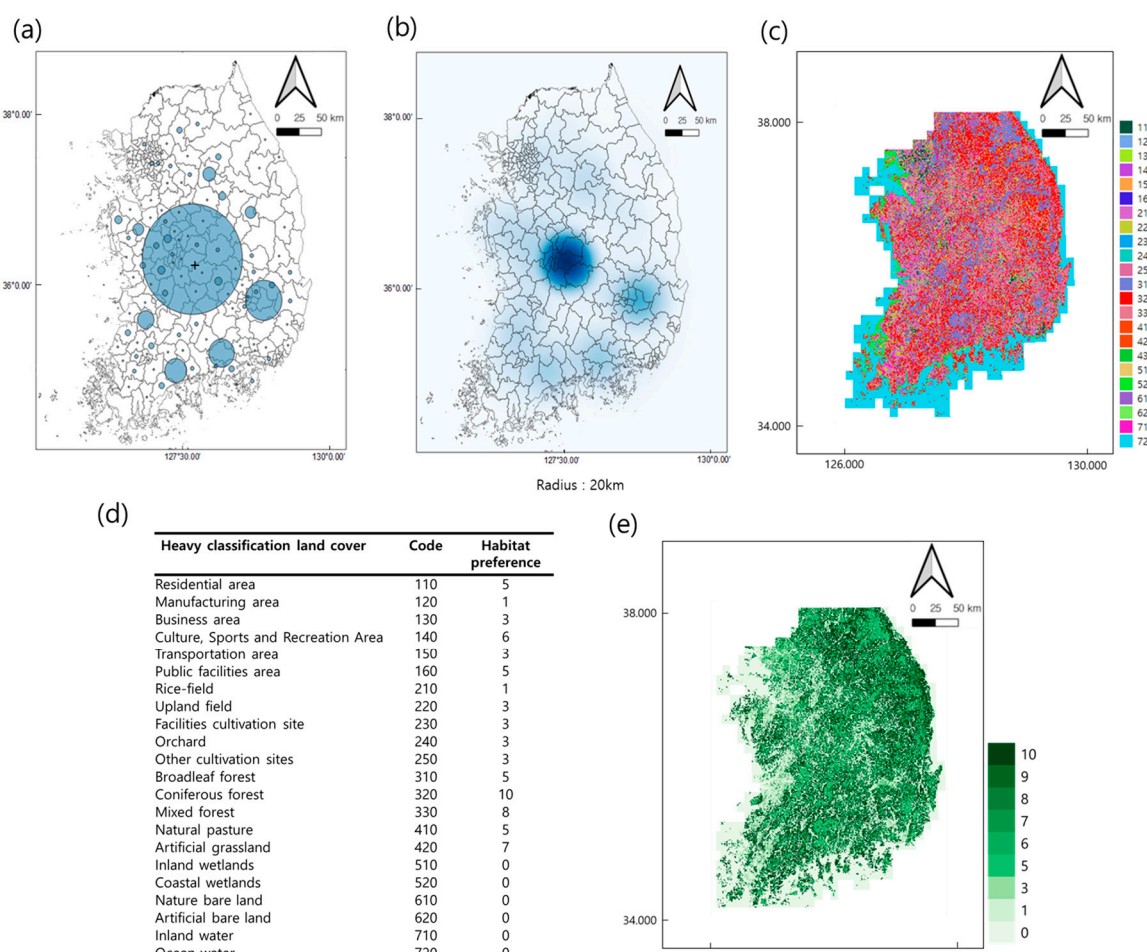

**Figure 3.** Distribution of sapling farms in Korea (blue circles presenting the number of sapling farms in local areas) (**a**), heat map (Radius = 20 km; stronger blue tones indicating higher densities) (**b**), map of land-cover types (**c**), habitat preference score for the WCSB (**d**), and spatial distribution of habitat preference on the map (**e**).

### 2.1.6. Habitat Preference

The direction of active movement of the WCSB was determined according to habitat preference. The land-cover data at the middle scale in Korea were downloaded from the Environmental Spatial Information Service (https://egis.me.go.kr/; accessed on 12 April 2022) (Figure 3c). Habitat preference was determined for land-cover in each spatial unit (1 km × 1 km) of the model (see Section 2.2.2). The level of habitat preference for each land-cover type was obtained according to the agreement of experts working on field surveys of the WCSB in Korea, which is similar to the empirical determination of resistance scores in land-cover types [26]. For each habitat type in each spatial unit, a number between 1 (least) and 10 (most) was assigned according to the judgment of the experts (Figure 3d). Figure 3e shows the overall distribution of habitat preference levels in the southern Korean peninsula.

### 2.2. Model Development
### 2.2.1. Overview

The models were developed to represent passive movements that link to the active movement of the WCSB. Passive movements were divided into traffic load and forest-product transport (See Section 2.2.4 for detailed information). A LSM was built to present population dynamics (i.e., growth and active movement) and accommodate passive movements because of traffic loads pertaining to local spatial units (Figure 4a). Subsequently, a SNM was constructed to simulate passive movement due to forest-production transport

between all local areas. SNM and LSM were linked in each iteration; transport of pests because of forest-product transportation obtained by SNM was combined with LSM at the corresponding locations (Figure 4b) (See Section 2.2.8).

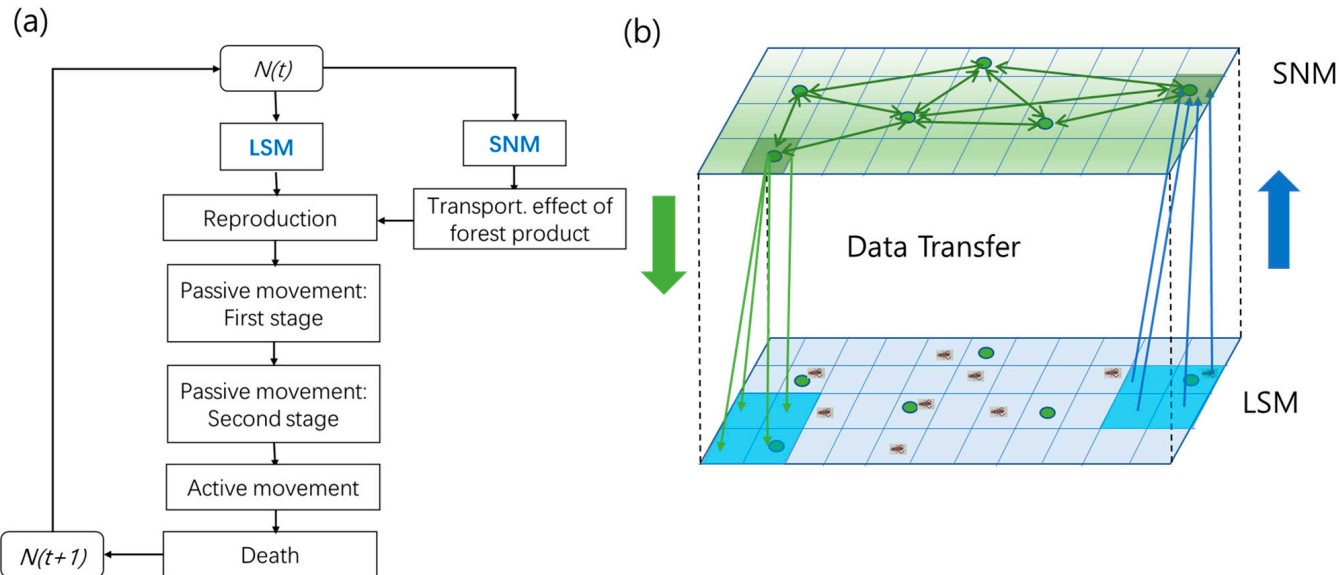

**Figure 4.** Incorporating lattice-structured model (LSM) and spatial network model (SNM) into simulating population dispersal of the WCSB: flow chart (**a**), and linkage graph of two models (green dots presenting imaginary positions of local areas, and blue and green arrows indicating data transfer from LSM to SNM and vice versa, respectively) (**b**).

### 2.2.2. System Definition

The model was constructed on the southern Korean peninsula (485 km × 344 km) with a unit of 1 km × 1 km in two dimensions for both LSM and SNM. The simulation was conducted for 20 years with a time unit of one year to present the yearly advancement of the pest population in recent years. The densities of the WCSB were provided as responding variables according to internal (e.g., Allee) and external (e.g., transportation) effects.

### 2.2.3. Life Events in LSM

Life events of the insect were presented at the adult stage in LSM, with movements occurring mainly at the adult stage [27]. Only females were considered for the simulation. Each female was assumed to produce 40 females [28]. Since only the adult stage was considered in the modeling, the model did not include metamorphosis for developmental stages according to the temperature change.

### 2.2.4. Passive Movement

In the passive movement, we considered two stages: reaching the nearest highway locally from the existing places of insects (Stage I) and long dispersal in the road network after reaching the highway (Stage II) (Figure 5a). Stage I was presented with the probability ($\theta x$) of arriving at the nearest road according to two variables: distance to the nearest road and traffic load ($d_1$ in Figure 5a). Stage II is expressed as the dispersal range ($d_2$ in Figure 5a) after reaching the highway network according to the traffic load.

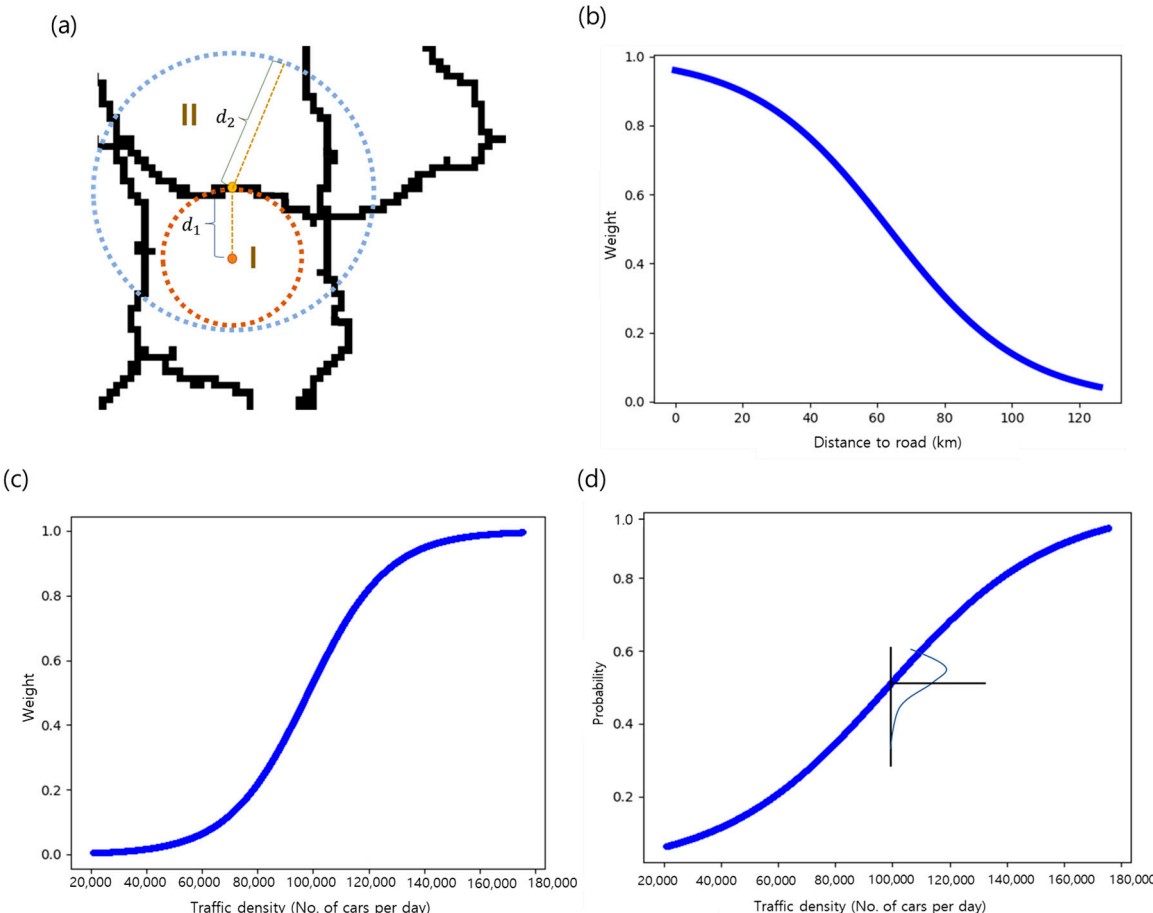

**Figure 5.** Two stages in passive movements in highway networks (The distance for the symbol $d_2$ is drawn intentionally short for simplicity of visualization). (**a**), probability of arriving at the road for Stage I according to distance to the road (**b**) and traffic load (**c**), and probability for dispersal distance for Stage II of passive movement according to traffic density (After the distance was determined by Equation (3), probability of dispersal in the model was generated according to the Poisson distribution) (**d**).

In Stage I, the density at each positional unit was transferred according to (1) the distance between the current position and the nearest point on the road (Figure 5b), and (2) the traffic density at the positional unit (Figure 5c). No known data are available regarding the effect of transport on insect movement. Logistic functions were assumed to determine the probability of arriving at the highway network, as shown in Equation (2) (Figure 5b,c). The values 63.5 km and 98,160 cars per day are the average distance to the nearest highway and the average traffic load, respectively.

$$\theta x = \frac{1}{1 + e^{-a_1 \times (d_x - 63.5)}} \times \frac{1}{1 + e^{-m_1 \times (T_x - 98,160)}} \tag{2}$$

$\theta x$: Rate of reaching the nearest point on the road from the current position.
$a_1$: Constant (0.05).
$d_x$: Maximum distance to the road (127 km) minus the distance to the road at the current position.
$m_1$: Constant (0.00007).
$T_x$: Traffic density at current position $x$.

After arriving at the road, the probability of dispersal was determined for Stage II in passive movement (Figure 5d). Dispersal probability was determined based on the traffic density from the position of arrival to the road network. This probability was

multiplied by the maximum distance to determine the dispersal range. The maximum dispersal distance ($d_m$ = 470 km, Table 1) in the road network was assumed to cover the entire southern Korean peninsula in one year (time unit in the model). The probability was determined according to the logistic function (Equation (3)). Once the distance was determined, the actual distances used in the model were generated again according to the Poisson distribution to provide randomness in the dispersal within the road network, using the value obtained from Equation (3) as the parameter for the Poisson distribution (Figure 5d).

$$P_m = \frac{1.0}{1 + e^{-m_2 \times (T_x - 98,160)}} \tag{3}$$

$P_m$: Probability of movement to the maximum distance (470 km).
$m_2$: Constant (0.000035).
$T_x$: Traffic load at the current position.

**Table 1.** Parameters and variables used for LSM and SNM (* Used for SNM, otherwise for LSM).

| Parameters | Description | Values | Sources |
|---|---|---|---|
| $r_d$ | Death rate | 0.7525 | [29] |
| $r_b$ | Number of progenies (female) produced per female | 40 | [28] |
| $K$ | Carrying capacity | 1,200,000/km$^2$ | Field experience |
| $A$ | Allee-effect threshold | 1000 | Tested in the model |
| $\gamma$ * | Contribution ratio of SNM | 0.25 | Tested in the model |
| $a_1$ | Slope for determining passive movement for Stage I according to distance to the road in the logistic function | 0.09 | Preliminary test |
| $m_1$ | Slope for determining passive movement for Stage I according to traffic load in the logistic function | 0.00007 | Preliminary test |
| $d_m$ | Maximum distance for dispersal in the road network for Stage I in passive movement | 470 km/year | Field data |
| $m_2$ | Slope for passive movement for Stage II according to traffic load in the logistic function | 0.000035 | Preliminary test |
| $b_1$ | Slope for active movement according to habitat preference difference in the logistic function | 0.09 | Preliminary test |

### 2.2.5. Movement Because of Forest-Product Transportation

As stated above, the transportation values of forest products between small local areas were obtained from the gravity rule (Equation (1)). Once the population densities in all spatial units were obtained using LSM, the data were transferred to the SNM. Population densities in different spatial units in the LSM were combined with the corresponding local areas in the SNM. Using these densities in local areas as node values and the amount of forest-product transportation between local areas obtained from the gravity rule as edge values, the total amount of incoming population densities to the target unit from all neighboring local areas (232 points) was calculated as follows: 1) the node and edge values were multiplied for each local area, and 2) these values for all neighboring areas were summed to be the WCSB densities for the target unit. The edge values were converted to a common logarithm before multiplication to reduce the effect of large values on the edges. A contribution ratio, $\gamma$ (unit; 1/tons), was separately given to this value before transferring the data to LSM to present suitable densities in LSM. In this study, the optimal value of $\gamma$ was verified through simulations.

### 2.2.6. Active Movement

Active movements consisted of two processes: determining the movement distance ($d$) and direction ($f(c)$), which were modeled in LSM along with Stage I of passive movement, as stated above (Figure 5a–c). The movement distance was obtained according to original data from field observations for adult flight over a period of 45 d [29] (Figure 6a). The negative exponential function (Equation (4)) was fitted to the field data. An extremely

high frequency was observed at a minimum distance of less than 2 km. Beyond 2 km, the frequency was very low (Figure 6a). Considering that the active movement period was approximately 180 days (six months) in the temperate region, the process of active movement was repeated four times in the model.

$$p = \frac{788.46927e^{-1.77538 \times d}}{325} \tag{4}$$

$p$: Probability of flight.
$d$: Flight distance.

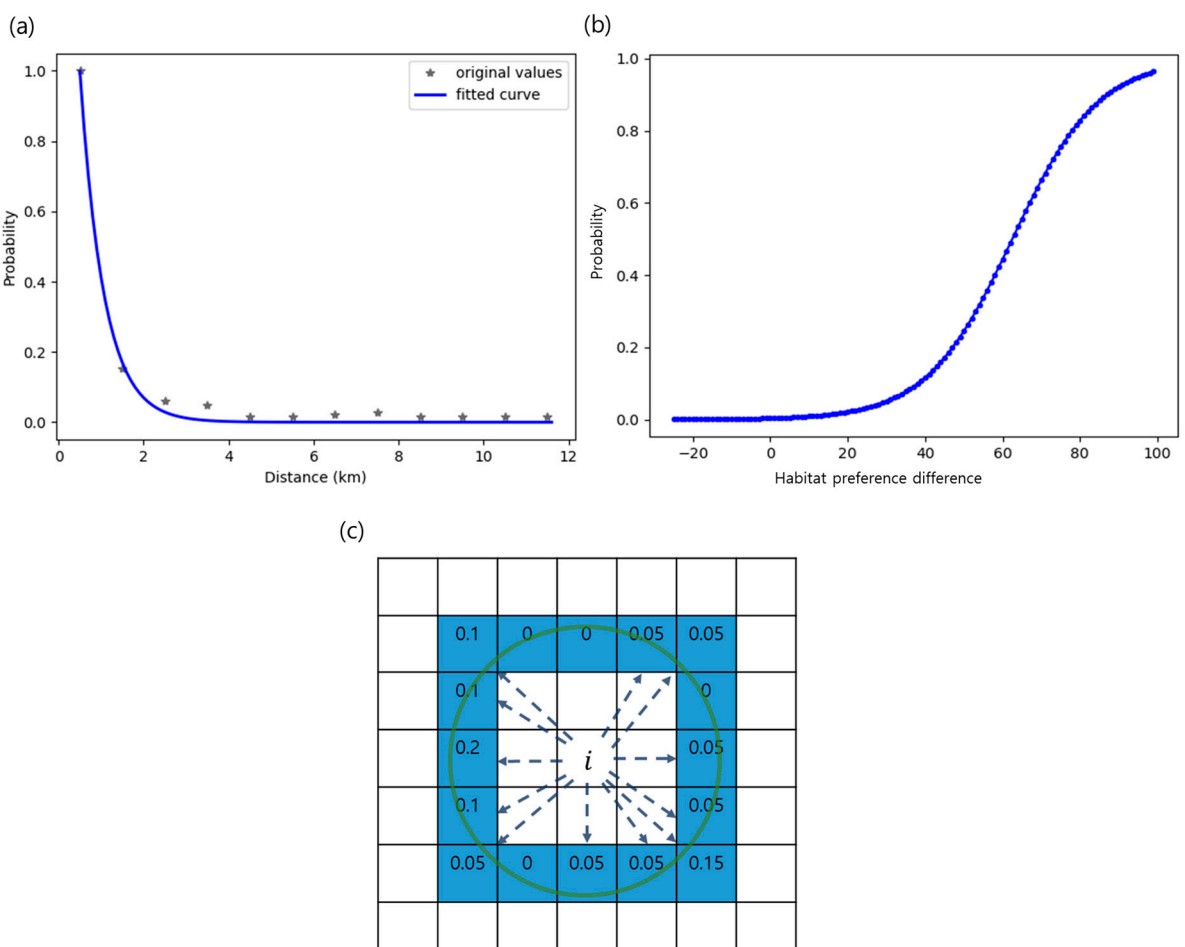

**Figure 6.** Determining probability of movement distance of the WCSB in the model according to field data (**a**), weight for movement direction based on habitat preference difference compared with the neighbor units (**b**), and probabilistic selection of spatial units according to habitat preference differences (dotted lines indicating probabilistic movement to new spatial units) (**c**).

The direction of active movement was determined using a logistic function according to differences in habitat preference (Figure 6b–c). From the target unit $i$, a circle was drawn with a radius for the dispersal distance based on Equation (4). Habitat preferences of all spatial units (Figure 3d) intersecting the perimeter of the circle were selected and habitat preferences in these units were used to calculate weights for movement direction ($f(c)$) according to Equation (5) (Figure 6b). The value 62.5 in the equation is the average value of habitat preference at location $i$ multiplied by the difference (absolute) between habitat preference at units $i$ and $j$. Subsequently, selection of spatial units was probabilistically

determined according to habitat preference differences (Figure 6c). Values higher than zero were only used for calculating weights and were subsequently normalized between 0.0–1.0.

$$f(c) = \frac{1}{1 + e^{-b_1(c - 62.5)}}$$ (5)

$f(c)$: Weight for determining the movement direction.
$b_1$: Constant (0.09).
$c$: Differences in habitat preference expressed as $c = p_j(p_j - p_i)$.
$p_j$ and $p_i$: Habitat preference (See Figure 3d) at cells $j$ and $i$ respectively, with movement from cell $i$ to cell $j$.

### 2.2.7. Population Dynamics

The densities of the pest population at time $t$ ($N_t$) were determined according to the equation listed below, based on the number of offspring ($r_b$), death rate ($r_d$), Allee-effect threshold ($A$), and carrying capacity ($K$).

$$\Delta N_t = r_d \times (r_b \times \frac{K - N_t}{K} \times \frac{N_t - A}{K})$$ (6)

$N_t$: Density of pest populations at time $t$.
$r_b$: Number of progenies (females) produced per adult female.
$r_d$: Death rate.
$K$: Carrying capacity.
$A$: Allee-effect threshold.

Only female adults were considered in the model and were assumed to produce 40 individuals per year ($r_b$) according to field data [28]. The values of $r_d$ and $K$ were 0.7525 [5] and 1,200,000 individuals/km$^2$ (in agreement with five field experts), respectively. Although the number of offspring was used to calculate population growth per year, the death rate was applied to the total population after all life events were completed each year in the simulation. The optimal value of the Allee-effect threshold $A$ was determined through a simulation in the range of 125–2000 individuals/km$^2$ after preliminary tests with the model.

### 2.2.8. Linking LSM and SNM

Given the initial conditions of the pest population, LSM simulated population dispersal according to rules related to population dynamics (i.e., growth and active movement) and Stage I of passive movement (Sections 2.2.4 and 2.2.6 and Section 2.2.7) (Figure 4a). The output data in the LSM spatial units were combined with the corresponding local areas in the SNM (Figure 4b). In SNM, the transported densities of the WCSB between local areas were calculated according to Section 2.2.5. The contribution ratio $\gamma$ was separately assigned to this value before transferring the data to LSM to present suitable densities in LSM, as stated above. Since the transported data were collectively obtained in each local area in the SNM, the data were dispersed into each spatial unit according to the heat map of sapling farms (Figure 3b) in LSM before the simulation proceeded in LSM. The simulation was repeated for the next iteration once the LSM received the data for population densities from the SNM (Figure 4b).

### 2.2.9. Parameters

The parameters used for modeling are summarized in Table 1. Most parameters are related to LSM, including growth, carrying capacity, and movement. The constant ($\gamma$) represents the contribution ratio of the SNM to the LSM. The Allee-effect threshold ($A$) and contribution ratio of SNM ($\gamma$) were the main parameters used to control population dispersal in this study based on preliminary tests. Suitable values for these parameters were determined according to changes in the parameter values in the simulation, as explained in the results (Section 3.1). The maximum distance ($d_m$) for dispersal in the road network for Stage I in the passive movement was 470 km, as explained in Section 2.2.4. The slopes in

the logistic functions, $m_1$, $m_2$, and $b_1$ for determining probabilities of dispersal in Stages I and II in passive movement (See Section 2.2.4) were obtained during preliminary tests. The parameter values for the slopes presented in Table 1 had stable ranges in responsive variables (spatiotemporal densities of WCSB) responding to different values of parameters $A$ and $\gamma$ in the simulation.

### 2.2.10. Initial Conditions

Initial populations of 30,000 female adults from the WCSB were randomly generated in the Changwon area (1 km$^2$) in 2010, ensuring population establishment with the initial density after preliminary tests.

### 2.2.11. Output Data

The spatially explicit density data are produced with an increase in iterations according to different parameters, including the Allee-effect threshold ($A$) and the constant controlling contribution by SNM ($\gamma$).

### 2.2.12. Stochasticity

Stochasticity was applied to the model through life events, including determination of the distance and direction of active movements, determination of arrival to the road in Stage I of passive movement, and distances of movement dispersal in Stage II of passive movement.

## 3. Results

### 3.1. Advancement Patterns

As shown in Figure 7, the combined LSM and SNM model presented fast, linear advancement of the WCSB populations along the highways according to the Allee-effect threshold, ($A$) equal to 1000 individuals/km$^2$ and a contribution ratio of SNM ($\gamma$) of 0.25. Based on the parameter values given to the model (Table 1), the simulation results were comparable to the field data (Figure 1). The top panel in each subfigure in Figure 7 shows an example of the overall population dispersal pattern on the map among 20 replicate simulations as time progressed from 2010 to 2030. Regarding spatial dispersal shown in the figure, the population occupied the southern area in 2015 (Figure 7b) and suddenly reached the Seoul area in the northwest area of the map in 2016 (dotted ellipse, Figure 7c). Jump dispersal was observed along the Gyeongbu Highway from Busan in the south to Seoul in the north (see the arrow in Figure 2d).

In the years following 2020, the population spanned the entire southern peninsula of Korea, starting with somewhat low densities with broad spatial dispersal in 2020 (Figure 7d). From 2022 onward, the populations were high in the southern Korean peninsula and became saturated afterward, as shown in Figure 7f. It should be noted that there were significant variations in the population densities in the simulation output. The figures show an example of spatial dispersal representing the simulated output optimally given by the parameter combinations (Table 1).

The bottom panel in each subfigure of Figure 7 shows the changes in the total population size obtained from LSM and SNM during the simulation between 2011 and 2030. The solid lines represent the average values of 20 simulations, whereas the shades show the range of density variation. Considering the high variation in population densities, the standard deviation (logarithm) was divided by the number of replicated simulations (20) to conveniently visualize the range of variation in the figures. Only the average is given for the first year of 2011 (bottom panel, Figure 7a). From 2015 onwards, the population started to grow rapidly. Notably, the total population level was higher with LSM than with SNM in the initial years until 2015 (Figure 7b). Subsequently, the population size obtained in SNM surpassed that obtained in LSM, matching the period when the population rapidly dispersed across the peninsula (arrows Figure 7b–e). Surpasses by SNM over LSM were observed in almost all cases when the population rapidly expanded in the simulation area. The population size tended to saturate around 12 years after introduction as shown in the

figure. If natural conditions are suitable for the survival of WCSB and there are no human effects (e.g., pest control), the trend of population advancement is predicted, as shown above, to reach saturation as early as the 2020s, approximately as early as 12 years after introduction.

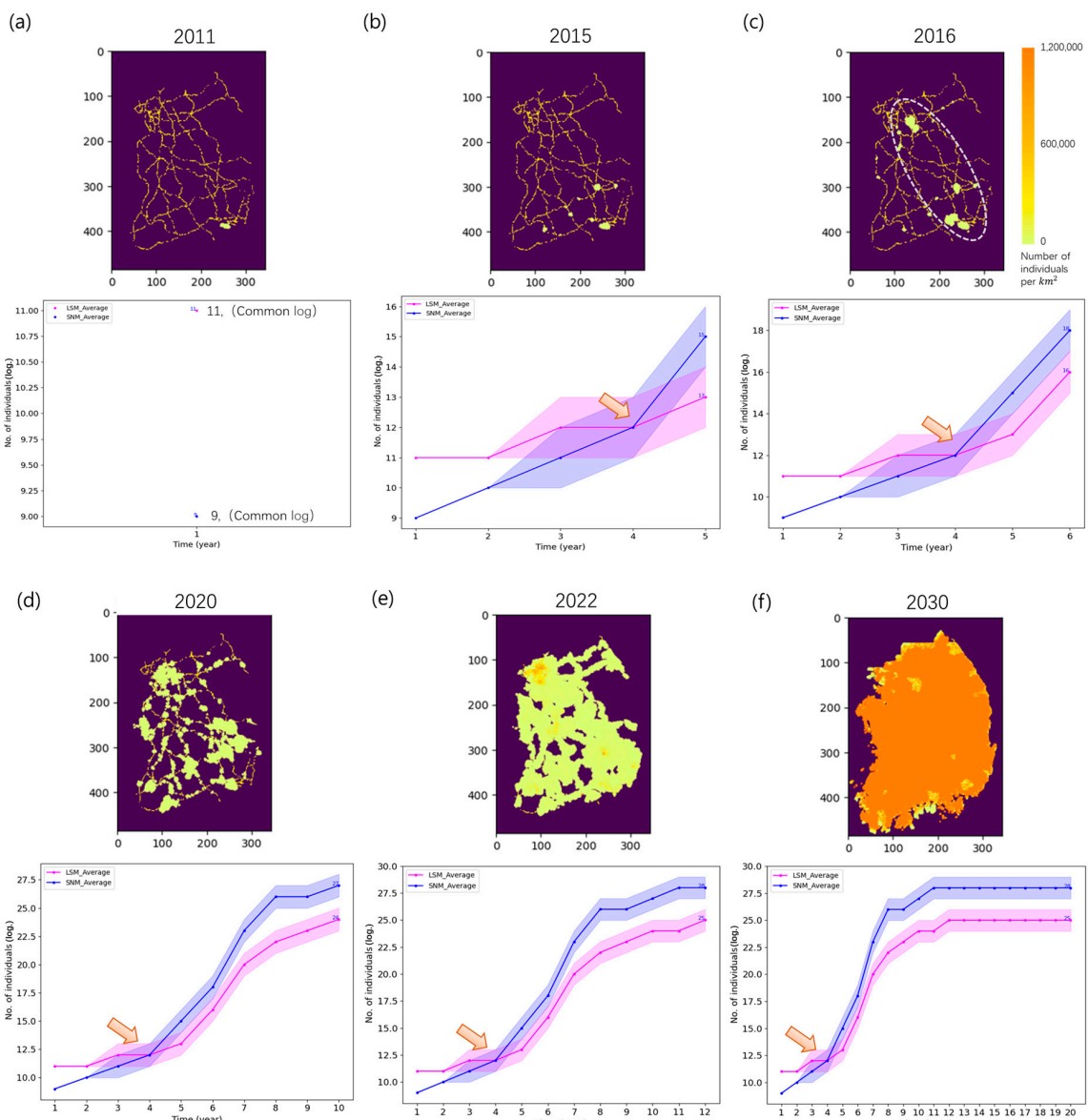

**Figure 7.** Simulating advancement patterns of WCSB populations between 2011–2030 in the southern Korean peninsula starting from 2010 according to LSM and SNM (Allee threshold; 1000 individuals/km$^2$, Gamma; 0.25) for the years of 2011 (**a**), 2015 (**b**), 2016 (**c**), 2020 (**d**), 2022 (**e**), and 2030 (**f**). In each subfigure top panel indicates spatial map and bottom panel presents increase in the total number of individuals (log.) during the simulation period. Horizontal and vertical units in two-dimensional maps in the top panel are expressed in km for convenience of illustration. Arrows in the bottom panel indicating the points where the population size obtained by SNM surpasses the population size obtained by LSM.

The overall population size on the Korean peninsula was estimated using LSM after receiving the densities from the SNM through a contribution ratio ($\gamma$ = 0.25) applied to forest-product transportation. The total number of insects reached 63,133,247,856 individuals in LSM and 1,681,331,234,162 individuals in SNM at the saturation level (2030, Figure 7f). The population level was higher in the SNM group than in the LSM group.

Output in LSM presented the total densities in field conditions after population densities from SNM (with conversion by $\gamma$) were transferred to LSM (See Section 2.2.8). The number of individuals (female adults) is hypothetical in the simulation but the model provided a possible quantitative range of individual numbers for survival in the southern Korean peninsula.

Population advancement was mainly controlled by the thresholds of the Allee-effect ($A$) and the contribution ratio of the SNM ($\gamma$). The Allee-effect threshold was suitable for the population dispersal of approximately 1000 individuals per square kilometer in this study. With low levels of the Allee-effect threshold and 125 individuals per unit area, population growth was faster and high levels of population density were reached earlier in 2020, as shown in Figure 8a,b. If the Allee-effect was set to a higher level of 2000 individuals per square kilometer, population advancement was delayed, occupying only the southeastern area of the peninsula (Figure 8c). However, saturation levels were also reached in 2030 in this case (Figure 8d).

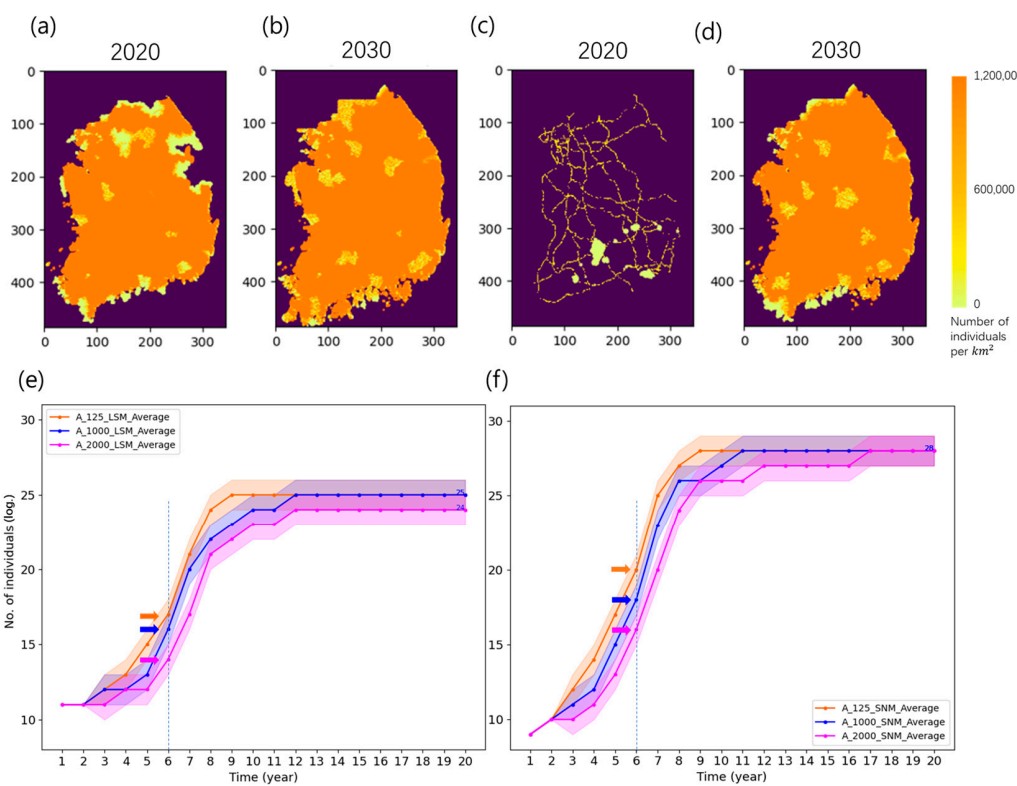

**Figure 8.** Simulating advancement patterns of WCSB populations with different Allee-effect thresholds ($A$) in the southern Korean peninsula in 2011–2030, with $A$ = 125 individuals per spatial unit in 2020 (**a**) and 2030 (**b**), and with $A$ = 2000 individuals per spatial unit in 2020 (**c**) and 2030 (**d**). (Horizontal and vertical units in two-dimensional maps are expressed in km for convenience of illustration.) Changes in population size (log.) of the WCSB with different $A$ ($\gamma$ = 0.25) with 125, 1000, and 2000 individuals per spatial unit in the southern Korean peninsula between 2010 and 2030 according to LSM (**e**) and SNM (**f**). Arrows in (**e**,**f**) indicating the number of individuals matching different levels of $A$ in the sixth year.

Figure 8e,f compare changes in total population size between LSM and SNM during the simulation between 2011 and 2030 with different Allee-effect thresholds of 125, 1000, and 2000 individuals per spatial unit. LSM and SNM populations grew and saturated around 12 years after the initial introduction, with overall levels higher in the SNM than in the LSM. It is noteworthy that population levels differed according to the levels of Allee-effect thresholds in the increasing phase of six years after the initial introduction, matching the time when the population rapidly reached the Seoul area. LSM population

levels had the highest level with the Allee-effect threshold of 125 individuals per spatial unit (orange arrow), followed by 1000 individuals per spatial unit (blue arrow), and 2000 individuals per spatial unit (red arrow) (Figure 8e). The difference in population size was also observed overall, similar to the threshold of the Allee-effect in the order of 125, 1000, and 2000 individuals per unit area. However, the difference range was wider in SNM than in LSM, as observed at high population sizes (Figure 8f). It is noteworthy that the difference in population size was greater at low Allee-effect thresholds between 125 and 1000 individuals/km$^2$ in SNM than in LSM (compare the two yellow arrows in Figure 8e,f) and early population growth because of low Allee-effect threshold levels was effectively illustrated globally in SNM. The Allee-effect controls the population size and spatial range in various ways during population dispersal.

The advancement pattern is also controlled by the contribution ratio ($\gamma$) of the SNM. We adjusted the ratio to 0.25 to present the WCSB according to forest-product transportation (Table 1). If the ratio was low at 0.15, the population advancement was spatially delayed; the insect population reached the Seoul area in 2020 (Figure 9a), whereas the population reached the Seoul area in 2016 with the same parameter condition with $\gamma$ equal to 0.25 (Figure 9c). However, with a low contribution ratio, population saturation was still observed in 2030 (Figure 9b). With higher levels of $\gamma$ equal to 0.35, the population expanded earlier, reaching substantially high levels over the peninsula in 2020 (Figure 9c), followed by saturation in 2030 (Figure 9d).

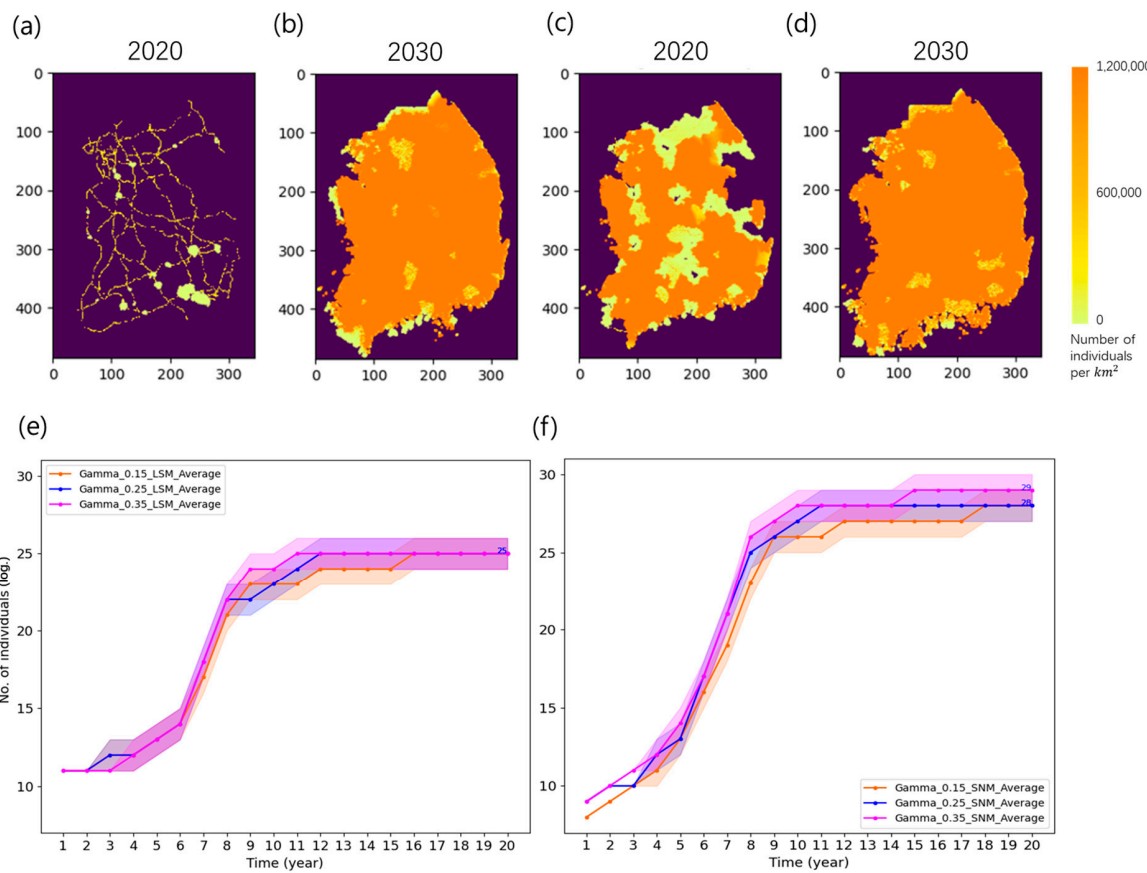

**Figure 9.** Simulating advancement patterns of WCSB populations with $\gamma$ = 0.15 in 2020 (**a**) and 2030 (**b**), and with $\gamma$ = 0.35 in 2020 (**c**) and 2030 (**d**) in the southern Korean peninsula starting from 2010 (Allee threshold; 1000 individuals per spatial unit). (Horizontal and vertical units in two-dimensional maps are expressed in km for convenience of illustration.) Changes in population size (log.) of WCSB with different conditions of $\gamma$ (0.15 and 0.35) in the southern Korean peninsula between 2011–2030, starting from 2010 according to LSM (**e**) and SNM (**f**).

Figure 9e,f compare the changes in the total population size obtained from LSM and SNM during the simulation between 2011 and 2030 with different contribution ratios of $\gamma$ (0.15, 0.25, and 0.35). It is noteworthy that population levels were invariable between different parameter conditions in the sixth year after introduction as the population rapidly expanded for both LSM (Figure 9e) and SNM (Figure 9f). This was different from the case of the Allee-effect, which showed relatively wider ranges in population size during this period (Figure 8e,f). This is understandable because SNM only contributes to the global transportation of the pest population between local areas and is not involved in local population dynamics, as presented in LSM. As expected, the overall densities were different between LSM (Figure 9e) and SNM (Figure 9f), with densities in SNM being higher than densities in LSM.

## 4. Discussion

Simulations in this study based on LSM and SNM in combination demonstrated fast, linear advancement of WCSB populations, possibly reaching saturation levels in the 2020s if the natural conditions remain favorable without human effects, including control efforts. Jump dispersal (dotted ellipses, Figure 10a,b) was substantially different from the conventional, slow, circular advancement, as shown in the case of invasion by the pine needle gall midge in Korea for a long period (Figure 10c) [15]. Busan, Mokpo, and Incheon, three initial points of invasion, produced circular advancement patterns and took around 40 years to spread over the southern Korean peninsula, in the case of the pine needle gall midge. Another example presents circular advancement within a city area (ca. 10 km) over a short period of approximately 10 years and still formed circular patterns in the case of the Pine wilt nematode (PWN) (Figure 10d).

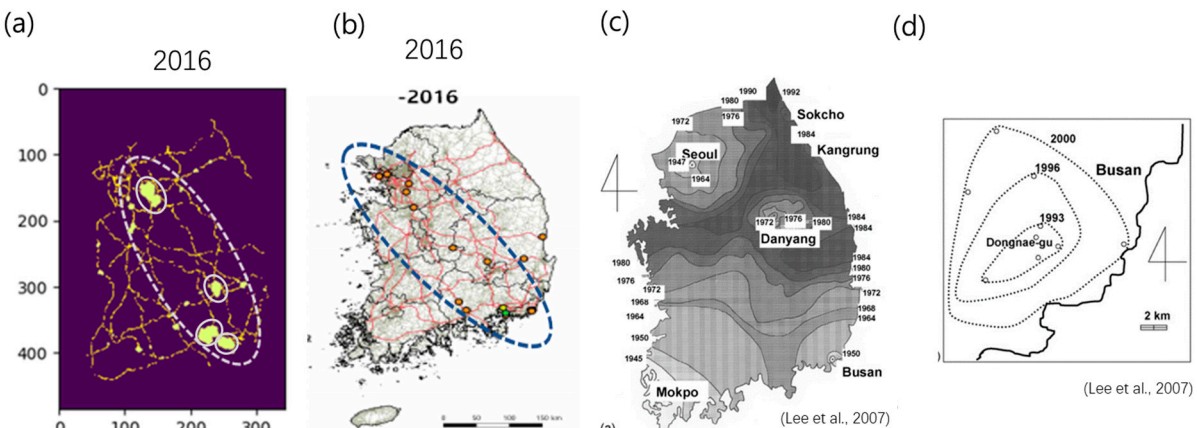

**Figure 10.** Simulation results for WCSB dispersal according to LSM plus SNM in 2016 (dotted ellipse indicating global, linear dispersal and solid ellipses presenting local, circular dispersal) (**a**), compared with field data (dotted ellipse indicating global, linear dispersal) (**b**), dispersal of Pine needle gall midge [15] (**c**), and dispersal of Pine wilt disease [15] (**d**). (The symbols resembling '4' in (**c,d**) present orientation with the top indicating north.)

It is noteworthy that rapid population advancement patterns were specifically visualized on the maps in this study. Passive movements with different processes were illustrated in the models. The passive movement because of the traffic effect presented the dispersal processes from the original places: the insects moved around from the source points (e.g., attaching to the cars). The forest-product transportation, however, had movement processes originating from other areas (i.e., sapling farms): the insects were transported between all local areas concurrently (Figure 2c). The difference in dispersal processes was effectively addressed in two models: the local traffic effect by LSM and global forest-product transportation by SNM.

Consequently, the LSM and SNM advancement patterns were characterized differently during the invasion. It was demonstrated that densities owing to SNM surpassed densities owing to LSM in the time for nationwide spreading (arrows, Figures 7 and 8), confirming a previous report suggesting the importance of sapling transportation [4]. The densities in SNM may also be high because of no constraint of carrying capacity given to SNM, contrary to the case of LSM, where carrying capacity limits population growth locally (Equation (6)). It is also noteworthy that circular movement, on a rather small scale, was produced according to LSM (solid ellipses in Figure 10a), demonstrating that LSM may present local dispersal according to population dynamics. Consequently, the combined model can effectively illustrate different human-mediated passive movements.

The effects of the parameters were also differentiated in simulation output. As shown in Figures 7–9, the population size was differentiated with different levels of Allee-effect thresholds (Figure 8e,f), whereas this differentiation was not clearly observed in the case of different contribution ratios ($\gamma$) of the SNM (Figure 9e,f). Although total densities were similar within a short range across different levels of Allee effects during the phase of increase in population size (Figure 8e,f), the ranges of total densities changed from the short range in the increasing phase to the wide range in the saturation point after around 10 years for the case of $\gamma$ value changes in both LSM (Figure 9e) and SNM (Figure 9f). This indicates that SNM may invariably contribute to fast increase in population size globally in the increasing phase, whereas LSM would govern local spatial dispersion patterns variably by the Allee-effect along with carrying capacity and spatial heterogeneity. Further research is warranted to investigate the differences between population growth and spatial dispersal in relation to SNM.

Widely distributed WCSB are harmful to needle-leaved trees, including the forest of Douglas fir (*Pseudotsuga menziesii*), mainly causing a decrease in seed production up to 41% [8,27]. In Korea, the pest causes economic damage to the production of Korean pine nuts (*Pinus koraiensis*) and may also decrease the density of Pinaceae trees, including the nationwide protected species, *Abies koreana* [3]. Additionally, aggregations at extremely high densities in human residency are often reported during winter [6]. Damage in various aspects, including natural (i.e., species conservation), agricultural (e.g., pine nut production), and human (e.g., aggregation in residency), are highly related to the rapid advancement of the WCSB populations. The current study focusing on passive movements of the pest would help devise control strategies, including preventing long-distance movement of forest-production (e.g., inspection of WCSB on trucks transporting saplings) and monitoring/cleaning of pests attached to cars moving in the areas of infestation.

A large range of variability was observed in population density in the simulation. Additional studies are required to address the variability of the model results, along with more replications in the simulation. The variability test could also be conducted with sensitivity analyses, including one-factor-at-a-time (OFAT) and Sobol tests [30], and could be studied in the future with high-performance computing systems.

In the simulation, the WCSB population was dispersed throughout Jeollanam-do in the early period including 2015 (Figures 7–9). The WCSB population, however, did not spread to Jeollanam-do in the west between 2010 and 2020, according to field data (Figure 1). This discrepancy may be because of a lack of environmental factors that might decrease dispersal from the eastern area to the western area. For instance, the mountain systems (Sobaeksanmak) that separated east and west would be one reason. The Allee-effect threshold pertaining to the western area may not be locally low enough to establish populations. Another reason may be the low degree of forest-product transportation between the eastern and western areas, as suggested in Figure 2c. The reason for this is currently unknown. However, further research is required in this regard.

The slopes in the logistic functions, $m_1$, $m_2$, and $b_1$ for determining probabilities of dispersal in Stages I and II in passive movement (see Section 2.2.4) were assumed to be constant in this study based on preliminary tests because the parameter values showed stable ranges in WCSB densities compared with substantial differences presented

by changes in the main parameters $A$ and $\gamma$ in the simulation. In the future, the contribution of these parameters to population densities should be further examined, along with more cases of field data.

A simulation time unit of one year was used in this model. Consequently, this model did not consider seasonal effects and metamorphosis (determining developmental stages from egg to adult in insects) according to temperature changes. This study mainly focused on demonstrating the fast, linear advancement pattern of WCSB populations because of human-mediated movements and other factors were simplified in the model. In the future, the time unit could be refined to months, weeks, or days for the model to incorporate temperature into determining insect development.

### 5. Conclusions

Model results according to LSM combined with SNM-matched field data between 2010 and 2020 presented linear jump dispersal because of the passive movement of the pest population along the highways in Korea. The model successfully simulated the fast populations spread from southeastern to northwestern Korea around 2016, possibly reaching saturation levels in the 2020s if the natural conditions remain favorable for the pest and no human control is conducted in the field. SNM effectively illustrated the global distance movement between local areas, whereas LSM presented population growth, active movement, and passive movements originating from current positions. Transportation according to forest products by SNM contributed to the wide dispersal of WCSB populations over LSM based on population dynamics and passive movement according to the traffic load.

The distance to the road, traffic load, and forest-product transport were effective in estimating the spatial dispersal of insect populations. The threshold of the Allee-effect and the transport of forest products play an important role in determining the dispersal of the insect population. Models combining SNM and LSM were feasible for presenting the human-mediated jump dispersal of forest pests.

**Author Contributions:** Conceptualization, T.-S.C.; methodology, X.Z., T.-S.C. and M.H.; software, X.Z.; data provision, D.-S.L., Y.-S.P., T.-G.L. and Y.-S.B.; validation and ecological evaluation, D.-S.L. and Y.-S.P.; and computational evaluation, I.-K.E. All authors have read and agreed to the published version of the manuscript.

**Funding:** This study was carried out with the support of the R&D Program for Forest Science Technology (Project No. 2017042B10-2323-CA01), provided by Korea Forest Service [22].

**Data Availability Statement:** Data for modeling are provided in the manuscript including parameters (Table 1). Availability of input/output data and source codes require permission from the funding organization.

**Acknowledgments:** We are indebted to Gyu-Suk Kwak and Yong-Hyeok Jang for providing input data for modeling and Uyyoon Park for checking the program codes.

**Conflicts of Interest:** The authors declare no conflict of interest. The funders had no role in the design of the study; in the collection, analyses, or interpretation of data; in the writing of the manuscript, or in the decision to publish the results.

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
