# Peer review of "Lattice Structure and Spatial Network Models Incorporating into Simulating Human-Mediated Dispersal of the Western Conifer Seed Bug Populations in South Korea"

_forests, doi:10.3390/f14030552_

Round 1
Reviewer 1 Report
Lack of line numbers makes it hard to comment.
Abstract – western should not be capitalized.
First paragraph – The vegetation or it’s condition is not contagious. The pathogen or pest is contagious. It can be “in spatially contiguous vegetation” or “in vegetation conditions that influence contagiousness.”
Expand first paragraph with basic life history of L. occidentalis. Also, the authors do not provide any justification as to why this is important to model. There is no economic or ecological context here - it spreads, that's it.
End of first paragraph. You cannot use the abbreviation WCSB without defining it first (abstract doesn’t count).
Second paragraph – “gypsy moth” is no longer the accepted common name for Lymantria dispar – it is spongy moth.
Sentence “Modeling of insect invasion...” needs to be reworded as it is awkward.
Figure 1 could be larger.
Section 1.1.2 – be consistent with naming – here it is Western seed bugs, but up to here, you’ve been calling it WCSB.
Figure 2 need proper legends (2a). Clarity is needed in text as to where the data comes from for Figure 2b and 2c. Sentence “...we obtained transportation between small areas...” is unclear.
Clarify the definition of highway in this data. Is there a traffic volume definition? Road width? Lane number? Why highway and not other provincial/state roads? Figure 3b labels it road, but is it truly highway?
Figure 13 – define the arrows. They are defined in the subsequent text, but need to be in the caption.
Figure 15 – same issue – arrows are defined in the text, but need to be defined in the caption.
Discussion - mostly restating the results. Very little in way of interpreting the results in the context of life history, economic or ecological impact related to spread, or other implications. It just spreads.
Author Response
"Please see the attachment."

Reviewer 2 Report
1. Fig 4b (Kriging data ) is not clear.
2. Fig 5 b have different scale from 5a. Both figures are depicting same data.
3. On what basis habitat preference was marked in Fig 6 b?
4. In many places values of constant are unclear.
5. Manuscript need proof reading
Author Response
"Please see the attachment."

Round 2
Reviewer 1 Report
I think the authors have clarified the justification and methods of the study. I still think the discussion is a bit thin, although the authors did add a paragraph to expand the economic implications of the pest. Overall, I think they did a sufficient job of addressing my concerns regarding the manuscript.
First paragraph – change “...America to Asia...” to “...North America to Asia...” unless the authors are combining North and South America, then it should be “...the Americas to Asia...”
Author Response
We appreciate greatly the reviewer for improving our manuscript. The contents and terminology in the manuscript were checked again, and the term commented by the reviewer was revised accordingly.
[line 42, first paragraph] from ‘America’ to ‘North America’